# Ethanol Reinforcement Elicits Novel Response Inhibition Behavior in a Rat Model of Ethanol Dependence

**DOI:** 10.3390/brainsci8070119

**Published:** 2018-06-26

**Authors:** Sucharita S. Somkuwar, Leon W. Quach, Jacqueline A. Quigley, Dvijen C. Purohit, McKenzie J. Fannon, George F. Koob, Chitra D. Mandyam

**Affiliations:** 1VA San Diego Healthcare System, San Diego, CA 92161, USA; lwquach@ucsd.edu (L.W.Q.); dpurohit@ucsd.edu (D.C.P.); mfannon@vapop.ucsd.edu (M.J.F.); 2The Scripps Research Institute, La Jolla, CA 92037, USA; jacqueline.a.quigley@gmail.com; 3National Institute on Drug Abuse, Baltimore, MD 21224, USA; george.koob@nih.gov; 4Department of Anesthesiology, University of California San Diego, San Diego, CA 92093, USA

**Keywords:** impulsivity, differential reinforcement of low rates, supersac, ethanol, CIE, dependence

## Abstract

Lower impulse control is a known risk factor for drug abuse vulnerability. Chronic experience with illicit drugs is suggested to enhance impulsivity and thereby perpetuate addiction. However, the nature of this relationship (directionality, causality) with regard to alcohol use disorder is unclear. The present study tested the hypothesis that higher impulsivity is observed during chronic intermittent ethanol vapor inhalation (CIE; a model of ethanol dependence) and subsequent abstinence from CIE in adult Wistar rats. Impulsivity was tested using a differential reinforcement of low rates 15 s (DRL15) schedule using either nondrug reward (palatable modified sucrose pellets) or sweetened ethanol. A decrease in the efficiency of earning reinforcers (expressed as % reinforcers/responses) is indicative of a decrease in response inhibition or an increase in impulsivity. The efficiency of reinforcement and amount of reinforcers earned were unaltered in CIE and control animals when the reinforcer was sucrose. When the reinforcer was sweetened ethanol, the efficiency of reinforcement increased in CIE rats compared with controls only during protracted abstinence. Responding for sweetened ethanol under a progressive-ratio schedule was more rapid in CIE rats during protracted abstinence. Contrary to the initial hypothesis, impulsivity did not increase in rats with a history of CIE; instead, it decreased when ethanol was used as the reinforcer. Furthermore, although the efficiency of ethanol reinforcement did not differ between CIE and control animals during CIE, CIE rats escalated the amount of sweetened ethanol consumed, suggesting that behavioral adaptations that are induced by CIE in rats that are tested under a DRL15 schedule appear to be targeted toward the maximization of ethanol intake and thus may contribute to escalation and relapse.

## 1. Introduction

Alcohol use disorder (AUD) afflicts approximately 17 million Americans [1]. The cluster of psychological symptoms that are associated with AUD broadly includes impairments in emotional regulation (e.g., higher aggression and anxiety), impairments in learning and memory, and executive function deficits [2,3,4,5]. Several of these conditions are indicators of impairments in prefrontal cortical function, a key brain region that regulates decision-making and impulse control [6,7,8]. Unsurprisingly, several lines of research reveal close associations between impulsivity and alcoholism [9,10]. For example, impulsivity has been suggested to be a predictor for future alcohol addiction vulnerability [11,12,13]. Furthermore, heavy alcohol use is suggested to trigger impulsive behavior [14,15,16]. Additionally, negative affect that is induced by chronic and heavy alcohol use may contribute to decreases in self-regulation and impulse control [17]. However, unclear is whether impulsivity enhances the risk for AUD or whether it is simply a co-occurring behavioral effect that is related to alternative predisposing (causal) factors that contribute to AUD. This causal relationship has been difficult to test, partially because of the heterogeneity of the impulsivity construct. Impulsive behaviors include actions that are poorly conceived, prematurely executed, unnecessarily risky, inappropriate, and often with undesirable outcomes (Evenden 1999). Understandably, these diverse phenotypes have overlapping but different etiological origins, and genetic and environmental factors may differentially affect each of these phenotypes. Therefore, investigating these phenotypes individually may yield a clearer understanding of their relationship with AUD.

Preclinical models have been immensely valuable, in which impulsivity is broadly divided into impulsive action (or lower response inhibition) and impulsive choice (choosing a less valuable reward that is immediately available over a more valuable reward that is available after a delay). Both of these aspects have been linked to addiction-like behavior in rodent models [18,19]. Alcohol-preferring (P) rats are a genetic model of excessive alcohol drinking. They exhibit innate impulsive choice [20] and response inhibition deficits [16,21,22,23]. However, response inhibition deficits were observed only when the cognitive load of the tests increased in such behavioral protocols as differential reinforcement of low rates (DRL) and the 5-choice serial reaction time test (5CSRTT) [16,22,23]. The present study investigated whether a history of excessive alcohol use alters impulsivity. Adolescence is a critical period in the maturation of the prefrontal cortex. Adolescent binge alcohol exposure (not adult binging) increased impulsive choice and decreased response inhibition during adulthood, and these effects were exacerbated by an acute ethanol challenge [16,24]. These results suggest that perturbation of the prefrontal cortex may lead to maladaptive impulse control. Unlike short binges, excessive alcohol exposure during adulthood, such as in chronic intermittent ethanol (CIE) paradigms, persistently alters the neurobiology of the prefrontal cortex [25,26,27,28,29,30]. Importantly, adaptations of cortical dopaminergic and pyramidal neuronal function that are produced by chronic ethanol [27,29,31,32] are similar to those that produce response inhibition deficits under a DRL schedule [33,34,35,36]. Therefore, we hypothesized that chronic ethanol vapor exposure (using CIE) impairs response inhibition under a DRL schedule. A within-subjects design was utilized to follow DRL responding over weeks of CIE and protracted abstinence. Particular emphasis was placed on Weeks 3–5 of abstinence, a critical time window for relapse vulnerability when some response inhibition deficits have been noted by others [27,37]. We also tested whether deficits in response inhibition are increased during protracted abstinence in rats with a history of CIE by enhancing the cognitive load via unpredictably increasing the required inhibition duration with DRL. Finally, DRL responding was reinforced using sweetened ethanol and a non-drug reinforcer (sucrose) to test the effect of the reinforcer on impulsivity.

## 2. Methods

### 2.1. Animals

Thirty-two adult male Wistar rats (Charles River; 8 weeks old upon arrival in the laboratory) were used for the study. The rats were pair-housed with *ad libitum* access to food and water under a reverse 12 h/12 h light/dark cycle. All of the experimental procedures were performed in strict adherence to the National Institutes of Health Guide for the Care and Use of Laboratory Animals (NIH publication No. 85–23, revised 1996) and were approved by the Institutional Animal Care and Use Committees at The Scripps Research Institute (protocol #07-0050-3) and VA San Diego Healthcare System (protocol #A16-000).

### 2.2. Materials and Apparatus

Differential reinforcement of low rate (DRL) experiments were conducted in operant conditioning modular chambers (ENV-001; MED Associates, St. Albans, VT, USA) that were housed in sound-attenuating compartments (ENV-018 M, MED Associates, St. Albans, VT, USA). A house light was mounted on the wall 7.3 cm above the metal grid floor.

For sucrose pellet-reinforced DRL schedules, these chambers were operated using ANYMaze Behavioral Tracking Software (version 6, Stoelting, Wood Dale, IL, USA) via an Ami-1 Interface. The chambers were equipped with an ANYMaze-compatible USB camera with varifocal lenses that enabled video tracking of the animals during the behavioral sessions. A 5 cm × 4.2 cm recessed food receptacle was installed on the wall that was opposite to the house light. Each receptacle was equipped with a built-in infrared beam and a sensor that detected nosepokes. Modified sucrose pellet reinforcers (Dustless Precision Pellet, 45 mg, Bio-Serv, Flemington, NJ, USA) were delivered via a dispenser (ENV-203, MED Associates, St. Albans, VT, USA) that was mounted outside the chamber.

For ethanol-reinforced DRL schedules, the chambers were operated using MED-PC software through a PC interface (SG-6080D, MED Associates, St. Albans, VT, USA). A sipper cup was mounted on the wall that was opposite to the house light, with a retractable lever on either side of the sipper cup. A single-speed single-syringe pump (PHM-100, Med Associates, St. Albans, VT, USA) was available in each operant chamber to deliver the ethanol solution (10% *w*/*v* ethanol in an aqueous medium of 1.5% *w*/*v* sucrose and 1.25% *w*/*v* saccharin sodium) from a plastic syringe to the sipper cup. This sweetened ethanol solution was similar to the previously described Supersac [38,39] and was selected to elicit high response rates that are not normally achievable using simply a 10% ethanol solution [25,26].

### 2.3. Chronic Intermittent Exposure Ethanol Vapor (CIE)

The ethanol dependence phenotype was generated by subjecting the rats to a widely validated regimen of CIE [25,40,41]. Specifically, the CIE model induced the escalation of voluntary ethanol intake, physical withdrawal, and negative affect-like responses that mimic AUD. During CIE, the rats were housed in specialized chambers and exposed to alcohol vapors on a 14-h ON/10-h OFF schedule in which alcohol (95% ethanol) was vaporized (by heating) and carried through the rat cages by controlled air flow (regulated by a pressure gauge). The rate of alcohol vaporization was regulated using a peristaltic pump (model QG-6, FMI Laboratory, Fluid Metering, Syosset, NY, USA) and optimized to produce blood ethanol concentrations (BECs) between 125 and 250 mg/dL (between 27.2 and 54.4 mM) [40]. These BECs are 2–3 times the BECs that are observed in binge drinking and are considered to be equivalent to 12–18 standard drinks in humans [42]. Notably, these BECs are not sufficiently high to abolish the righting reflex [43,44]. During the first day of CIE, low ethanol vapor levels were applied. The levels were then gradually increased each day during the first week to prevent overexposure in the previously ethanol-naive animals [40]. Blood ethanol levels were determined only once weekly to avoid the undue stress that can be caused by repeated blood collection. This procedure of gradually increasing ethanol exposure over the first week has been previously used to generate a rodent model of ethanol dependence [25,40,41].

### 2.4. Blood Ethanol Concentrations

Blood ethanol concentrations were measured weekly (every Friday) between 13 and 14 h of vapor exposure [45]. Briefly, the rats were gently restrained while the tip of the tail was pricked with a clean needle. Tail blood (0.2 mL) was collected in specialized anticoagulant-coated microcentrifuge tubes (Fischer Scientific, Hannover Park, IL, USA) and centrifuged at 2000 rotations per minute for 10 min. Plasma (5 μL) was used to measure BECs using an Analox AM1 analyzer (Analox Instruments, Lunenburg, MA, USA). Single-point calibrations were performed for each set of samples with reagents that were provided by Analox Instruments (100 mg/dL). Plasma from CIE-naive control rats was used to determine “instrument/technical noise,” and the mean of this error value (3.9 ± 0.92) was deducted from the BEC of rats that underwent CIE to obtain actual BECs. When BECs were outside the target range (125–250 mg/dL), vapor levels were adjusted accordingly.

### 2.5. Differential Reinforcement of Low-Rate of Responding

All behavioral testing was conducted during the dark cycle. CIE rats (*n* = 16) and control rats (*n* = 16) were trained on a fixed-ratio 1 schedule (30 min/session) to nosepoke in the food receptacle to obtain a single modified sucrose pellet (*n* = 16; *n* = 8 CIE and *n* = 8 controls) or to press the active lever (the retractable levers remained extended during the operant sessions) to obtain 100 mL of ethanol solution (*n* = 16; *n* = 8 CIE and *n* = 8 controls). The rats were then introduced to the DRL schedule [46,47]. Initially, the minimum time between consecutive responses (interresponse time (IRT)) to obtain a reinforcer was 2 s (DRL2). The rats were trained from DRL2 to DRL 15 (i.e., reinforcement of IRT > 15 s) in daily 30-min sessions by gradually increasing the schedule requirement by 0.75% within-sessions following each reinforced response. If DRL15 was not achieved in a single session, then the adjusted DRL interval was carried over to the next session until DRL15 was reached [35,46]. DRL15 was selected because a previous study showed that responding by adult Wistar rats exhibited ceiling effects at DRL5 and floor effects by DRL30 [35]. The rats continued to be tested on DRL15 for 5 days per week until stable performance was reached, defined as <10% variation in efficiency over 3 consecutive sessions. The rats were divided into groups that were subjected to CIE and groups that were not subjected to CIE (control), thus forming the following groups: SUCROSE-CIE, SUCROSE-CON, ETHANOL-CIE, and ETHANOL-CON (*n* = 8/group). DRL15 testing continued for 1 week prevapor (Week 0), six weeks of CIE (Weeks 1–6), and the subsequent 5 weeks of forced abstinence from CIE (Weeks 7–11). Data were collected 5 days per week, but data from Monday and Friday were excluded to avoid confounds that could be caused by a lack of testing over the weekends and because of blood draw-related stress, respectively, although separate analyses of operant behavior on Mondays and Fridays did not reveal any evident trend or differences from the other days (data not shown). Notably, DRL15 sessions were conducted 5–7 h into the vapor-OFF period during Weeks 1–6. CIE rats during this phase exhibit physical withdrawal, the escalation of voluntary ethanol drinking, and an increase in anxiety-like behavior, all signs of a negative emotional state [26,48,49]. During the last 3 weeks of withdrawal, all of the rats underwent a one-session challenge in which the reinforcement contingency was abruptly switched to DRL20 to test whether rats with a history of CIE respond differently than controls when the cognitive load is enhanced.

### 2.6. Progressive-Ratio Schedule

Following the last DRL session (i.e., Week 11), the ETHANOL-CIE and ETHANOL-CON groups were tested for responding on a progressive-ratio schedule to determine the reinforcing efficacy and/or motivation to obtain ethanol [50]. In a progressive-ratio schedule, the effort or “cost” (i.e., the number of lever presses) that is required for a single reinforcer progressively increases until the subject stops responding for a 15-min period (i.e., the breakpoint) [51,52]. As such, progressive-ratio breakpoints are suggested to be a measure of compulsivity [53]. The pattern of lever pressing under this schedule was plotted as a cumulative distribution curve and compared between groups. Greater motivation may lead to impairments in response inhibition and thus may affect DRL performance. Therefore, we assessed the possible contribution of motivation to any group differences in DRL responding.

### 2.7. Locomotor Activity

The video tracking algorithm in the ANYMaze system was utilized to determine the distance travelled in sucrose rats (measured using the center of the animal as a reference point) during the DRL15 session. The average distance travelled over three sessions during Prevapor, during Week 5 (acute withdrawal), and during Week 11 (5 weeks of abstinence) was compared between CIE rats and control rats. Locomotor activity was not measured in ethanol rats because video tracking was unavailable here, and ethanol intake during DRL15 may confound the data interpretation.

## 3. Data Analysis

### 3.1. Mathematical Analysis of Interresponse Time Data: Modified Temporal Regulation Model

This analysis was conducted as a complementary analysis of IRT distribution to better characterize behavior under DRL15 schedules. Although several different approaches can be used to characterize this type of IRT data [35,46,54,55,56], we present the analysis using a modified temporal regulation (TR) model. This modified TR model (described below; Equation (1)) assumes that responding under DRL15 are a composite of a gamma distribution pattern (not normal or Gaussian) for responses that are centered around the 15 s schedule requirement (referred to as time-responses for DRL15) and exponential decay functions for other, perhaps nonspecific responses (referred to as non-timed responses) [35,46,54]. The modified TR model was applied to the cumulative frequency distribution of IRTs using Microsoft Excel software, and parameters were obtained for each rat after maximizing the likelihood of fit.
(1)$$P\left(IRT=t\right)=p\Gamma \left(t-\delta ;N,c\right)+q\left(1-p\right)Le^{-L\left(t-\delta \right)}+\left(1-q\right)\left(1-p\right)L^{\prime }e^{-L^{\prime }\left(t-\delta \right)}$$

The first component of the above-modified TR model equation is the gamma distribution, followed by the two exponential decay functions (*e^x^*) for the nonspecific short IRTs (burst) and very long IRTs that were elicited by each subject under DRL15. The following parameters from mathematical models served as dependent variables to describe response inhibition capacity:Response threshold θ = (*N* × *c*)/15, where *N* and *c* are the shape and scale parameters for the gamma distribution. (*N* × *c*) is the mean of timed IRTs. θ < 1 indicates lower accuracy of timed IRTs.The proportion of timed IRTs (*p*) is expressed as a fraction of all IRTs that were obtained from an individual rat. A larger *p* value indicates greater response inhibition capacity.The proportion of burst IRTs (*q* × (1 – *p*)) is expressed as a fraction of all IRTs that were obtained from an individual rat. Burst IRTs are a component of non-timed IRTs. A greater proportion of burst IRTs indicates lower response inhibition capacity.Rate-of-decay, *L*, which is the exponent fitting the burst IRTs under DRL15. A steeper decay may result from a larger drop in burst responding.

The model also provides the Weber-fraction (ω), an estimation of the precision of IRTs. The smallest IRT that was elicited was δ. The fraction ((1 – *p*) × (1 – *q*)) and rate-of-decay (*L’*) of long IRTs under DRL15 characterize the very long IRTs that may indicate distraction from the task or task-delinquent responses [57].

### 3.2. Parametric Data

One rat in the SUCROSE-CON group died at Week 6 of CIE. Post-mortem analysis did not reveal any detectable cause of death. A technical malfunction resulted in the sporadic loss of DRL data for one rat in the ETHANOL-CIE group and one rat in the ETHANOL-CON group. Therefore, data for these three rats are missing for all the repeated-measures analyses and for the selective parametric data analysis. Although data were collected 5 days per week, only data from Tuesday–Thursday were used to obtain weekly averages. Data that were collected during Monday and Friday were excluded to remove any confounds due to lack of testing over the weekends, and due to blood-draw related stress, even though separate analyses for operant behavior on Mondays and Fridays did not reveal any particular trend or differences from the other days (data not presented). No comparisons were conducted between the different types of reinforcers because of the following differences: (*i*) physical properties of the reinforcer (solid vs. solution), (*ii*) response type (nosepoke vs. lever press), and (*iii*) position of the manipulandum (at the reinforcement site vs. adjacent to the reinforcement site). Therefore, our primary comparisons were between rats with and without CIE experience and time. All of the data are reported as mean ± SEM. The number of rats per group (*n*) is reported. The data were analyzed using two-way repeated-measures analysis of variance (ANOVA) using GraphPad Prism software (version 7, GraphPad Inc, San Diego, CA, USA). Time (in weeks, 12 levels) and time-bins (0.5 s intervals, 120 levels) were within-subjects factors, and history of CIE (CIE and CON, 2 levels) was the between-subjects factor. The efficiency of reinforcement (% responses reinforced), number of sucrose or ethanol reinforcers earned, locomotor activity, lever responding under the progressive-ratio schedule, the IRT frequency distribution, and TR modeling-dependent variables for response inhibition were compared between groups. Because the latter weeks of prolonged abstinence are important from the perspective of relapse vulnerability [26,27], behavior during Week 11 was investigated in greater detail using separate *t*-tests to compare CIE and control animals. Significant interactions and main effects were followed by the Sidak post hoc test, corrected for multiple comparisons. The cumulative frequency distribution for responding under the progressive-ratio schedule was compared between the ETHANOL-CIE and ETHANOL-CON groups using the Kolmogorov–Smirnov test. The significance was assessed at α = 0.05.

## 4. Results

Blood ethanol concentrations in CIE rats were successfully maintained within the desired range between Weeks 2 and 6 (Figure 1a and Figure 2a). The BECs for Week 1 were below the desired range, but this was an expected consequence (see detail in Methods section) of the gradual increase in vapor exposure during the first week. The BEC data were not analyzed during Week 5 in the ETHANOL-CIE group because of a malfunction of the Analox Instrument. All of the groups exhibited increases in body weight over weeks of CIE (Table 1). Baseline differences in body weight persisted through the weeks of behavior testing.

### 4.1. Reinforcement Efficacy of Sucrose during Acute Withdrawal and Prolonged Abstinence from CIE

Response inhibition, measured as the efficiency of obtaining sucrose reinforcer, was not different between the SUCROSE-CIE and SUCROSE-CON (ethanol-naive) groups during the 6 weeks of vapor exposure and 5 weeks of abstinence (Figure 1b; *F*_interaction_(11, 143) = 1.33, *p* = 0.21; *F*_CIEhistory_(1, 13) = 0.39, *p* = 0.54). A significant main effect of time on reinforcement efficiency was observed (*F*_time_(11, 143) = 7.82, *p* < 0.0001). Post hoc comparisons revealed that reinforcement efficiency at Weeks 9–11 (abstinence Weeks 3–5) was greater than at Prevapor (*p* < 0.02; Figure 1b). These results suggest that reinforcement efficiency under DRL15 increased over weeks of DRL experience, but a history of CIE *per se* did not affect it.

Significant main effects of time and CIE on the number of sucrose pellets that were earned during CIE and abstinence were observed (*F*_time_(11, 143) = 2.17, *p* = 0.02; *F*_CIEhistory_(1, 13) = 9.10, *p* = 0.01), but no interaction was found (*F*_interaction_(11, 143) = 0.64, *p* = 0.79). The greater number of pellets that were earned in the SUCROSE-CON group appears to reflect cohort-to-cohort variation, such that the SUCROSE-CON group earned a greater number of pellets compared with the SUCROSE-CIE group during Prevapor (*t*(14) = 2.50, *p* = 0.03; Figure 1c), and this baseline difference persisted through weeks of CIE and abstinence. Further analysis of the main effect of time using Sidak’s post hoc test separately for the SUCROSE-CIE and SUCROSE-CON groups did not reveal any significant pairwise differences, suggesting that neither DRL experience nor CIE history affected the number of sucrose reinforcers that were earned in either group.

### 4.2. Interresponse Time Distribution and Mathematical Modeling Data for Rats that Responded to Sucrose Reinforcers

Separate *t*-tests of reinforcement efficiency at Week 11 with a sucrose reinforcer did not reveal an effect of CIE history (*t*(13) = 0.55, *p* = 0.50; Figure 3). The frequency distribution for the sucrose-reinforced DRL revealed a significant effect of IRT bin (*F*_IRTbin_(119, 1547) = 39.08, *p* < 0.0001) but no effect of CIE history and no interaction (*F*_CIEhistory_(1, 13) = 2.91, *p* = 0.11; *F*_interaction_(119, 1547) = 0.40, *p* > 0.9999). Mathematical modeling of the IRT data did not reveal any effect of CIE history (Table 2), which is consistent with the IRT frequency distribution results.

### 4.3. Reinforcement Efficacy with Ethanol during Acute Withdrawal and Prolonged Abstinence

The efficiency of obtaining an ethanol reinforcer was not different between the ETHANOL-CIE and ETHANOL-CON groups during the 6 weeks of vapor exposure and 5 weeks of abstinence (*F*_interaction_(11, 132) = 1.54, *p* = 0.12; *F*_CIEhistory_(1, 12) = 2.17, *p* = 0.17; Figure 2b). A significant main effect of time was observed (*F*_time_(11, 132) = 5.71, *p* < 0.0001), such that reinforcement efficiency increased at Weeks 5–11 (i.e., CIE Weeks 5–6 and abstinence Weeks 1–5) compared with Prevapor (*p* < 0.05). These results replicate findings from the sucrose-reinforced DRL15, in which the reinforcement efficiency for ethanol also increased over weeks of DRL experience. Separate *t*-tests of reinforcement efficiency at Week 11 revealed that ethanol increased reinforcement efficiency in rats with a history of CIE (*t*(12) = 2.27, *p* = 0.042; Figure 2b). Although the effect size was small (Cohen’s *d* = 1.22, *r* = 0.52), these findings are opposite to the expected results that reinforcement efficiency, a measure of response inhibition capacity (i.e., an inverse indicator of impulsivity), would be reduced by a history of CIE.

Ethanol intake during CIE and abstinence (number of ethanol reinforcers earned and ethanol consumed) showed a clear main effect of time (*F*_time_(12, 144) = 6.51, *p* < 0.0001; Figure 2c,d) and a time × CIE history interaction (*F*_interaction_(12, 144) = 2.97, *p* = 0.0001) but no main effect of CIE history (*F*_CIEhistory_(1, 12) = 6.99, *p* = 0.02). *Post hoc* comparisons revealed that ethanol intake at Week 5 was only greater in the ETHANOL-CIE group compared with the ETHANOL-CON group (*p* = 0.012). Compared with the respective Prevapor ethanol intake, the ETHANOL-CIE group exhibited an increase in ethanol consumption at Weeks 1–6 (i.e., six weeks of vapor; *p* < 0.01; Figure 2d) and at Weeks 8–9 (i.e., Weeks 2–3 of abstinence; *p* < 0.01). In contrast, the ETHANOL-CON group do not exhibit changes in ethanol intake over weeks of testing.

### 4.4. Interresponse Time Distribution and Mathematical Modeling Data for Rats that Responded to Ethanol Reinforcers

Further investigation of the increase in reinforcement efficiency was conducted by analyzing the frequency distribution of IRTs and mathematical data-fitting to obtain distribution parameters [47,54]. The frequency distribution for ethanol-reinforced DRL15 revealed a significant main effect of IRT time bin and a significant IRT time bin × CIE history interaction (*F*_IRTbin_(119, 1428) = 55.86, *p* < 0.0001; *F*_interaction_(119, 1428) = 2.43, *p* < 0.0001; Figure 4) but no effect of CIE history (*F*_CIEhistory_(1, 12) = 0.05, *p* = 0.83). The significant effect of time bin was an expected reinforcement schedule characteristic. Further investigation of the interaction revealed that the ETHANOL-CIE and ETHANOL-CON groups differed in the proportion of burst IRTs (IRT time bins 0.5–1.5 s; *p* < 0.05) and in the proportion of timed IRTs that are just shy of the 15 s reinforcement cut-off (IRT time bins 12–13.5 s; *p* < 0.05; Figure 4). Mathematical modeling of IRT data supported the observed differences in burst IRTs such that the rate of decay of burst responding was lower in the ETHANOL-CIE group compared with the ETHANOL-CON group (*t*(132) = 4.34, *p* < 0.0001; Figure 4), but no other parameters were different between groups (Table 2). A lower rate of decay is consistent with the lower proportion of short IRTs that was observed in the ETHANOL-CIE group (Figure 4). The lower frequency of short bursts and rightward shift of timed IRTs are indicative of greater response inhibition or lower impulsivity in the ETHANOL-CIE group.

### 4.5. Response Inhibition and Reinforcer Intake during an Unexpected DRL Challenge

The cognitive load for successfully obtaining the reinforcers was increased by changing the reinforcement contingency to DRL20 for one randomly selected session during prolonged abstinence (Week 10) to test whether rats with a history of CIE adapt their behavior differently than the corresponding CIE-naive control group. Regardless of the reinforcer that was used, the efficiency of responding decreased during the challenge session compared with the week’s average efficiency under DRL15 (*F*_challenge_(1, 13) = 93.84, *p* < 0.0001, and *F*_challenge_(11, 12) = 40.34, *p* < 0.0001, for sucrose and ethanol, respectively; Figure 5a and Figure 6a). No effect of CIE history and no DRL challenge × CIE history interaction were found for either sucrose (*F*_CIEhistory_(1, 13) = 0.66, *p* = 0.43; *F*_interaction_(1, 13) = 1.13, *p* = 0.41) or ethanol (*F*_CIEhistory_(1, 12) = 1.94, *p* = 0.19; *F*_interaction_(1, 12) = 0.12, *p* = 0.73) reinforced schedules (Figure 5b and Figure 6b).

### 4.6. Progressive-Ratio Schedule Using Ethanol Reinforcer

After the last DRL15 session, the ETHANOL-CIE and ETHANOL-CON groups were tested on a progressive-ratio schedule to identify differences in efficacy of the ethanol reinforcer or motivation to obtain the ethanol reinforcer. No differences were observed in the number of lever responses or the number of ethanol reinforcers that were earned (*t*(13) = 0.30, *p* = 0.77, *t*(13) = 0.30, *p* = 0.77, respectively; Table 3). Further analysis of active lever responses revealed significant group differences in the cumulative distributions (Kolmogorov–Smirnov *D* = 0.617, *p* < 0.0001; Figure 7), in which the ETHANOL-CIE group exhibited a steeper rise earlier for the ethanol reinforcer compared with the ETHANOL-CON group. However, responding plateaus were comparable by the end of the session for both groups.

### 4.7. Locomotor Activity during DRL Responding

Locomotor activity was not different between rats with a history of CIE and ethanol-naive rats that were responding for the sucrose reinforcer during prevapor, Weeks 5 and 11 (*F*_interaction_(2, 41) = 0.08, *p* = 0.92; *F*_CIEhistory_(1, 41) = 0.60, *p* = 0.44; *F*_time_(2, 41) = 1.08, *p* = 0.35; Table 4).

## 5. Discussion

The novel findings in the present study were that CIE rats modified their responding under DRL schedules to earn reinforcers more efficiently during protracted abstinence, but no evidence of lower efficiency was observed during CIE or during abstinence. Several different methodological manipulations were utilized to evaluate whether rats with a history of CIE exhibit deficits in response inhibition. However, the data consistently refuted this hypothesis. Altogether, these data provide abundant evidence that exposure to excessive ethanol did not induce response inhibition deficits. This apparent divergence of results from clinical observations of the exacerbation of impulse control deficits in chronic alcoholics [14,15,16] is perhaps attributable to the absence of preexisting impulse control deficits in outbred Wistar rats. High impulsivity in clinical subjects with heavy alcohol use may be attributable to the ethanol-mediated exacerbation of preexisting impulse control deficits and not manifest in non-impulsive subjects after heavy alcohol consumption. This hypothesis may be tested in preclinical models of high impulsivity and subsequent exposure to CIE followed by testing behavioral adaptations over weeks of CIE and abstinence. The greater efficiency of earning ethanol reinforcement during protracted abstinence may be a behavioral adaptation that enables the maximization of reinforcement. Further evidence of adaptations to maximize ethanol reinforcement was the increase in the amount of ethanol reinforcers that were earned during acute withdrawal and prolonged abstinence. Furthermore, a steeper increase in responding under a progressive-ratio schedule was associated with an improvement of response inhibition for ethanol. These data contrast with several studies that suggested that the greater motivation toward drugs of abuse contributes to greater deficits in response inhibition. In fact, the increase in efficiency at Week 11 in the ETHANOL-CIE group suggests an improvement of response inhibition capacity at a time-point when previous studies reported greater relapse vulnerability [26,27]. We hypothesize that when the reinforcer is a drug of abuse, such as ethanol, the motivation for the drug contributes to improvements of response inhibition to facilitate the escalation of drinking and increase relapse vulnerability. Future studies should investigate whether these results with ethanol are generalizable to other drugs of abuse.

These behavioral adaptations were absent when a natural reinforcer (i.e., sucrose) was used. No evidence of impairments in or improvements of response inhibition was observed. No differences in sucrose intake were observed between the SUCROSE-CIE group and CIE-naive controls. Sucrose-reinforced operant responding (i.e., instrumental conditioning in which the strength of behavior is modified by reinforcement) [58] that was utilized herein was considerably different from ethanol-reinforced operant responding, which precluded direct comparisons between the two. However, when the effects were compared between rats with and without a history of CIE within each reinforcement condition, these data clearly challenged the widely accepted notion that escalation and relapse in ethanol dependence are attributable to impairments in response inhibition that are induced by excessive ethanol exposure. In fact, no trace of response inhibition deficits was observed at Week 11 in the SUCROSE-CIE group, a time-point when previous studies reported greater relapse vulnerability in rats with a history of CIE [26,27]. To date, only one study has reported a decrease in response inhibition during ethanol abstinence. However, this effect was transient and not replicated over repeated cycles of CIE and abstinence [37]. The continuous testing protocol that was used herein may have masked such small and transient effects. Additionally, Irimia et al. [37] tested CIE-abstinent rats under conditions of partial food restriction (i.e., food availability reduced to 90% of free-feeding body weight) and not under conditions of *ad libitum* access to food and water that were utilized herein. Thus, the interpretation of response inhibition in the present study was not confounded by the possibility of a food/water-deprivation effect. In fact, the present results support previous findings that a history of chronic ethanol exposure during adulthood did not alter impulsivity in adult subjects [23,24]. Although other impulse-control deficits may be increased by a history of excessive ethanol experience, the present results strongly argue against the possibility that an increase in response inhibition deficits contributes to an increase in ethanol relapse in the CIE model of ethanol dependence.

In the present study, the number of ethanol reinforcers that were earned increased from ~37 at prevapor to ~64 at Weeks 5 and 6 of vapor exposure, whereas the number of sucrose reinforcers that were earned by CIE rats varied between 60 and 75 for the duration of testing. One limitation of the current interpretation is that the number of sucrose pellets that were earned may be at or near a ceiling of responding within the confines of the 30-min session, and this may obfuscate the effect of CIE on the intake of the sucrose reinforcer. However, the absence of an effect with sucrose and the escalation of ethanol intake in the ETHANOL-CIE group may also be explained by higher tolerance to the acute effects of ethanol in rats with chronic ethanol experience. A previous study showed that the oral intake of 0.8–1.0 g/kg ethanol in ethanol-naive Wistar rats produced BECs of 60–80 mg/dL [59]. Comparable BECs, particularly in rodents without a genetic predisposition for excessive ethanol intake, are associated with arousal, such as greater dopamine release in the nucleus accumbens [59] and a slight increase in locomotor activity [60,61,62]. In contrast, higher ethanol intake in such rodents decreased locomotor activity [60,61,62], decreased risk-taking behavior, and decreased learning and memory performance [62,63,64]. Although tolerance to the oral intake of 1.0–1.5 g/kg ethanol is not likely attributable to changes in pharmacokinetics [65], the possibility of psychodynamic tolerance cannot necessarily be excluded. Thus, prolonged exposure to high levels of ethanol may increase tolerance to ethanol’s rewarding effects, drive negative reinforcement and effect greater compulsive-like responding as measured by increased responding for a bitter-tasting quinine-alcohol solution [66], and thus may underlie the escalation of ethanol intake that was observed in the present study.

Prefrontal cortex deficits under DRL schedules typically manifest as an increase in impulsivity (lower reinforcement efficiency), some cortical deficits may manifest a different phenotype. For example, lesions of the ventrolateral frontal cortex resulted in an increase in reinforcement efficiency under a DRL schedule [67], that suggest “hoarding” or “optimal foraging”. Thus, the increase in efficiency that was observed in the ETHANOL-CIE group and the associated reduction of burst responding and rightward shift of the timed-responding peak may be associated with such behavior. Alternatively, DRL with ethanol reinforcers may tap into motivation- and/or cue reactivity-related neurocircuits that are critical in the transition from recreational to addictive drug use [68,69]. These behaviors are also linked to prefrontal cortical deficits [14,28,70,71,72]. This is important because the persistent escalation of voluntary ethanol intake is a hallmark of such a transition to addictive drug use that is observed in several models of AUD in adult subjects [26,41,48,73,74,75,76,77]. Notably, the escalation of ethanol intake was observed under the more challenging operant paradigm and was obtained through an increase in the number of responses without compromising reinforcement efficiency (i.e., no increase in premature responding). Furthermore, this effort to maximize ethanol reward may reflect hypothesized reward deficit- and stress surfeit-driven negative reinforcement in CIE rats that was previously reported as compulsive-like alcohol seeking [17,28]. A combination of conditioned cues (e.g., operant chambers) and interoceptive cues (e.g., physiological effects of ethanol) may produce the motivation to increase intake to compensate for the lower pharmacological efficacy of ethanol in CIE subjects.

Although progressive-ratio responding for sucrose was not evaluated in the present study, others have reported that food-motivated operant responding, including progressive-ratio responding, was unaltered in rats that were exposed to other treatments with CIE [78], suggesting that these changes are selective to the drug of abuse. Indeed, greater motivation in rodent models of alcohol dependence has been demonstrated by an increase in progressive-ratio breakpoints for 10% *w*/*v* ethanol during acute withdrawal [75,76,79] and during protracted abstinence [76]. The present study revealed that although overall motivation indices were unaltered, a leftward and upward shift of the cumulative progressive-ratio responding curve was observed in the ETHANOL-CIE group. The ETHANOL-CON group exhibited a pause in responding after 20 min into the schedule, but they otherwise continued to respond at a slow but steady rate under the progressive-ratio schedule. The progressive-ratio schedule that was utilized herein was more challenging than the one that was used by Vendruscolo et al. [76], which may explain the absence of a higher breakpoint or number of reinforcers that were earned in the ETHANOL-CIE group. The more stringent schedule was adopted to compensate for the assumed increase in the palatability of the ethanol reinforcer that was caused by the addition of sucrose and saccharin [38,39]. Stress-mediated alterations of hedonic states in CIE rats may have contributed to the early plateau of progressive-ratio responding in CIE rats [39,48,80,81,82,83] and to the greater efficiency of DRL responding [84]. Finally, the association between lower impulsivity with higher progressive-ratio responding was quite unique because greater motivation is often associated with greater impulsivity. This further supports the hypothesis that response inhibition behavior is an adaptation that supports an increase in ethanol intake in rats with a history of CIE. Altogether, these data suggest that the ETHANOL-CIE group exhibited greater motivation for or tolerance to ethanol, which was reflected by an increase in ethanol intake under DRL.

One limitation of the present study may be that response inhibition with the DRL15 schedule was not stable across the duration of testing and exhibited an increase with excessive training. Specifically, regardless of the reinforcer type, responding contingency, and history of CIE, all of the rats exhibited an increase in reinforcement efficiency after 5–10 weeks of repeated DRL15 testing. Typically, in rodent behavioral studies, stability is defined over much shorter time intervals (e.g., a couple of days). In the present study, stability was defined as <10% variation in the mean efficiency of reinforcement, which was achieved and a precondition for the initiation of CIE. Testing continued for several weeks (almost 3 months) after the first evidence of stability, and this shift in efficiency is thus not a surprising or remarkable outcome of repeated testing. Another factor may be the age of the rats because age, among other factors, is known to alter response inhibition, even in humans [85,86].

## 6. Conclusions 

In conclusion, the present study found that impulsivity, contrary to our original hypothesis, measured as deficits in reinforcement efficacy on a DRL15 schedule, did not increase during acute withdrawal or prolonged abstinence from excessive ethanol exposure. Studies of impulsivity in models of AUD suggest that chronic ethanol exposure and protracted abstinence have only very mild effects on this measure of response inhibition. Thus, preexisting response inhibition deficits and perhaps their exacerbation following excessive ethanol intake may contribute to the escalation of intake and increase in relapse risk in individuals with AUD. Greater motivation for ethanol or an increase in tolerance to the acute effects of alcohol may reflect psychopharmacological effects of prolonged exposure to high levels of ethanol that may synergistically contribute to the escalation of intake and increase in relapse risk.

## Figures and Tables

**Figure 1 brainsci-08-00119-f001:**
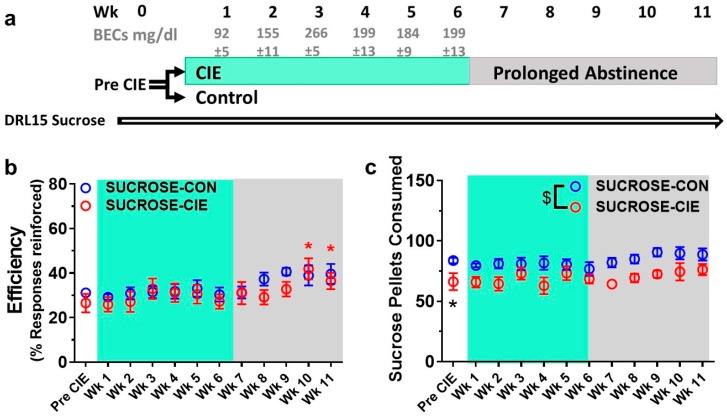
Response inhibition for obtaining sucrose reinforcement is unaltered during protracted abstinence from excessive alcohol exposure. (**a**) Experimental paradigm. Rats were trained until they achieved stable responding on a DRL15 schedule (Pre CIE) using sucrose reinforcement. The rats continued to be tested on the same schedule 5 days/week for 6 weeks (CIE, chronic intermittent ethanol, Weeks 1–6; BECs are reported for each week during CIE) during acute withdrawal from ethanol vapor or air exposure and subsequently for 5 weeks (protracted abstinence, Weeks 7–11). (**b**) Efficiency of sucrose reinforcement (open circles) increased at Weeks 9–11 compared with Pre CIE but was not different between the SUCROSE-CIE and SUCROSE-CON groups at Week 11. (**c**) Baseline differences in sucrose intake on DRL15 was found at Pre CIE. The data are expressed as mean ± SEM for percent reinforcers earned/responses emitted. *n* = 7–8/group. * *p* < 0.05, compared with Pre CIE in (**b**); * *p* < 0.05 compared to SUCROSE-CON in (**c**); ^$^
*p* < 0.05, significant main effect of CIE in (**c**). CIE: chronic intermittent ethanol; DRL: differential reinforcement of low rates.

**Figure 2 brainsci-08-00119-f002:**
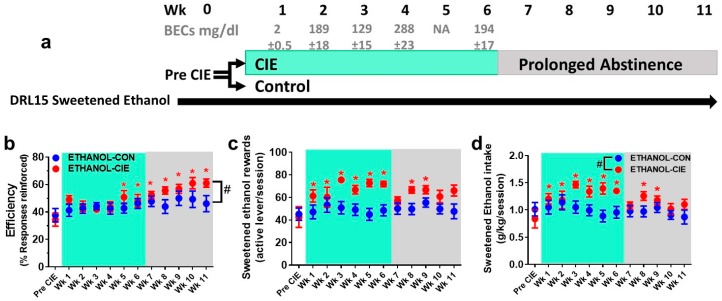
Response inhibition for obtaining ethanol reinforcement increased during protracted abstinence from excessive ethanol exposure. (**a**) Experimental paradigm. The rats were trained until they achieved stable responding on a DRL15 schedule (Pre CIE) using ethanol reinforcement (10% ethanol; 0.1 mL per reward). The rats continued to be tested on the same schedule 5 days/week for 6 weeks (CIE, chronic intermittent ethanol, Weeks 1–6; BECs, blood ethanol concentrations, are reported for each week during CIE; NA, not available) during acute withdrawal from ethanol vapor or air exposure and subsequently for 5 weeks (protracted abstinence, Weeks 7–11). (**b**) Efficiency for ethanol reinforcement (closed circles) increased at Weeks 5–11 compared with Pre CIE and was greater in the ETHANOL-CIE group compared with the ETHANOL-CON group. (**c**,**d**) An increase in ethanol intake was observed during acute withdrawal and protracted abstinence from CIE. Acute withdrawal increased ethanol intake compared with the ETHANOL-CON group during Weeks 1–6. A significant increase in intake was observed in the ETHANOL-CIE group compared with the ETHANOL-CON group at Week 5. This increase in ethanol intake compared with pre CIE continued into Weeks 8–9 of protracted abstinence. The data are expressed as the mean ± SEM for percent reinforcers earned/responses emitted. *n* = 7–8/group. * *p* < 0.05, compared with Pre CIE in (**b**–**d**); # *p* < 0.05, compared with ETHANOL-CON group in (**b**); # *p* < 0.05, significant interaction and applies to (**c**,**d**).

**Figure 3 brainsci-08-00119-f003:**
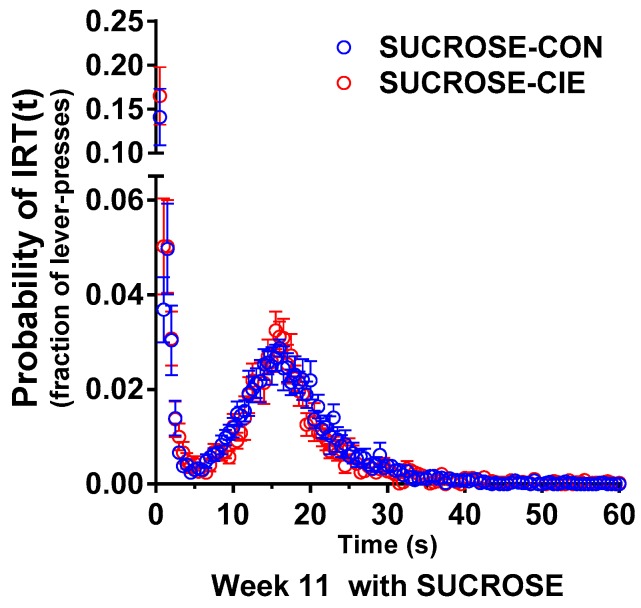
Interresponse time (IRT) distribution during DRL15 at Week 11 was unaffected in the SUCROSE-CIE group. No difference in the IRT distribution between the SUCROSE-CIE group (open red circles) and SUCROSE-CON group (open blue circles) was observed at Week 11. The data are expressed as mean ± SEM for percent reinforcers earned/responses emitted. *n* = 7–8/group.

**Figure 4 brainsci-08-00119-f004:**
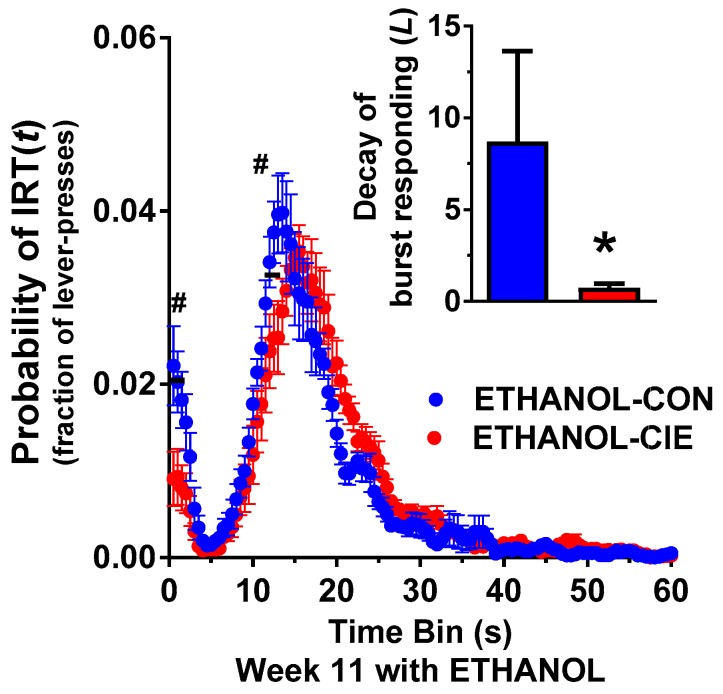
Interresponse time (IRT) distribution during DRL15 at Week 11 was modified in the ETHANOL-CIE group, such that response inhibition increased compared with the ETHANOL-CON group. (Inset) Mathematical modeling revealed that the ETHANOL-CIE group (red) exhibited a smaller rate of decay of burst responding compared with the ETHANOL-CON group (blue). The data are expressed as the mean ± SEM for percent reinforcers earned/responses emitted. *n* = 7–8/group; ^#^
*p* < 0.05, compared with IRT in ETHANOL-CON group at corresponding time bin; * *p* < 0.05, compared with corresponding CON group (inset).

**Figure 5 brainsci-08-00119-f005:**
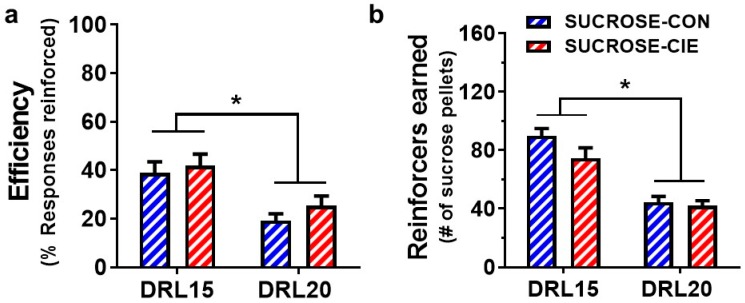
Efficiency of reinforcement and reinforcer intake were reduced during an unexpected DRL20 schedule compared with under the DRL15 schedule in sucrose animals. No differences were found between the SUCROSE-CIE group (red) and SUCROSE-CON group (blue) in (**a**) the efficiency of sucrose reinforcement or (**b**) the number of sucrose reinforcers that were earned. The data are expressed as mean ± SEM for percent reinforcers earned/responses emitted. *n* = 7–8/group. * *p* < 0.05, compared with corresponding values during DRL20 challenge.

**Figure 6 brainsci-08-00119-f006:**
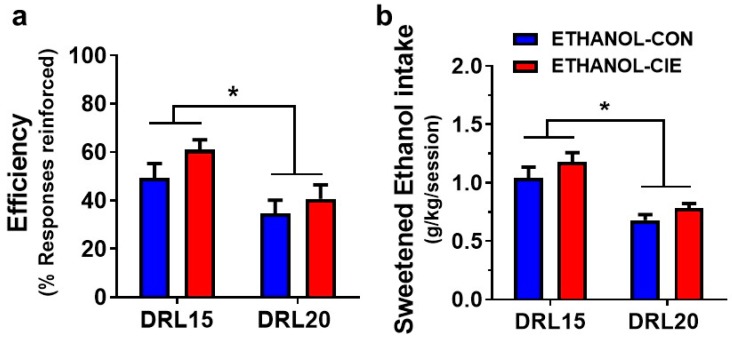
Efficiency of reinforcement and reinforcer intake decreased during an unexpected DRL20 schedule compared with under the DRL15 schedule in ethanol animals. No differences were found between the ETHANOL-CIE group (red) and ETHANOL-CON group (blue) in (**a**) the efficiency of ethanol reinforcement or (**b**) the number of sucrose reinforcers earned. The data are expressed as the mean ± SEM for percent reinforcers earned/responses emitted. *n* = 7–8/group. * *p* < 0.05, compared with corresponding values during the DRL20 challenge.

**Figure 7 brainsci-08-00119-f007:**
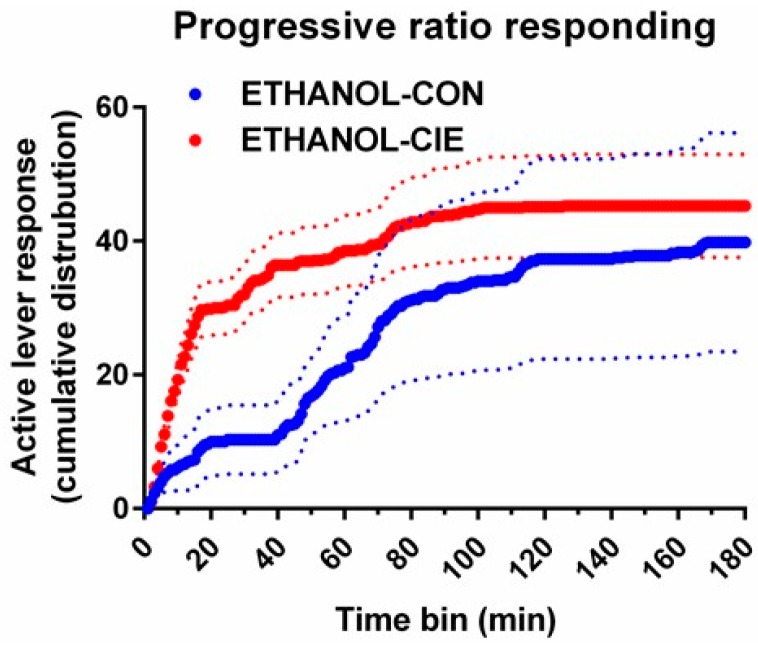
The ETHANOL-CIE group exhibited greater motivation for the ethanol reinforcer during protracted withdrawal compared with controls. This effect was reflected by a steeper rise in lever responding in the ETHANOL-CIE group (red) compared with the ETHANOL-CON group (blue). The data are expressed as the mean ± SEM for cumulative active lever responding that was emitted under the progressive-ratio schedule. *n* = 8/group.

**Table 1 brainsci-08-00119-t001:** Body weights of rats did not differ between groups.

GROUPS	SUCROSE-CON	SUCROSE-CIE	ETHANOL-CON	ETHANOL-CIE
**Prevapor**	447 ± 12.4	411 ± 6.36	393 ± 6.84	434 ± 9.07
**Week 1 (Vapor 1)**	470 ± 13.0	462 ± 8.11	407 ± 7.02	457 ± 12.1
**Week 2 (Vapor 2)**	NA	477 ± 8.50	421 ± 7.87	464 ± 14.1
**Week 3 (Vapor 3)**	516 ± 15.2	493 ± 10.3	429 ± 8.89	478 ± 13.2
**Week 4 (Vapor 4)**	500 ± 21.5	507 ± 12.1	449 ± 9.29	506 ± 15.2
**Week 5 (Vapor 5)**	531 ± 15.1	523 ± 12.3	470 ± 9.60	507 ± 16.1
**Week 6 (Vapor 6)**	555 ± 12.8 ^#^	NA	488 ± 10.2	517 ± 15.3
**Week 7 (Abstinence 1)**	569 ± 13.1 ^#^	548 ± 13.5	499 ± 10.9	505 ± 19.1
**Week 8 (Abstinence 2)**	581 ± 12.8 ^#^	563 ± 15.4	510 ± 11.8	525 ± 16.7
**Week 9 (Abstinence 3)**	597 ± 15.7 ^#^	585 ± 16.6	506 ± 13.9	534 ± 17.4
**Week 10 (Abstinence 4)**	601 ± 15.1 ^#^	589 ± 17.8	540 ± 14.4	570 ± 18.4
**Week 11 (Abstinence 5)**	612 ± 14.6 ^#^	606 ± 20.2	555 ± 14.1	601 ± 19.6

The data are in grams and expressed as mean ± SEM. *n* = 8/group (except ^#^
*n* = 7/group). NA, not available.

**Table 2 brainsci-08-00119-t002:** Mathematical modeling parameters for sucrose-reinforced DRL responding.

Groups	Responses	Response Distribution	CON	CIE	t Ratio
**SUCROSE**	Proportion of timed responses	Proportion gamma; *p*	0.67 ± 0.04	0.57 ± 0.07	*t*(143) = 0.05
	Accuracy of timed responses	Response threshold; θ = (*N* × *c*)/15	1.17 ± 0.05	1.11 ± 0.03	*t*(143) = 0.03
	Proportion of burst responses	Proportion short exponent; (*q* × (1 – *p*))	0.16 ± .03	0.29 ± .07	*t*(143) = 0.07
	Rate of decay of burst responses	Rate of decay for short exponent *L*	5.80 ± 2.22	4.42 ± 2.86	*t*(143) = 0.77
**ETHANOL**	Proportion of timed responses	Proportion gamma; *p*	0.78 ± 0.05	0.77 ± 2.86	*t*(132) = 0.00
	Accuracy of timed responses	Response threshold; θ = (*N* × *c*)/15	1.12 ± 0.11	1.07 ± 0.10	*t*(132) = 0.03
	Proportion of burst responses	Proportion short exponent; (*q* × (1 – *p*))	0.07 ± 0.02	0.11 ± 0.04	*t*(132) = 0.02

The data are expressed as mean ± SEM. *n* = 7–8/group.

**Table 3 brainsci-08-00119-t003:** Progressive-ratio breakpoints and number of ethanol reinforcers earned were not different between groups.

Reinforcers Earned	ETHANOL-CON	ETHANOL-CIE
**Lever responses**	33.6 ± 14.8	38.5 ± 8.55
**Ethanol reinforcers earned**	4.00 ± 1.33	5.25 ± 0.88

The data are expressed as mean ± SEM. *n* = 7–8/group.

**Table 4 brainsci-08-00119-t004:** Locomotor activity (in meters) did not differ between CIE and control rats.

Group	Prevapor (Week 0)	Acute Withdrawal (Week 5)	Prolonged Abstinence (Week 11)
**SUCROSE-CON**	65.0 ± 14.8	86.3 ± 12.9	50.5 ± 6.17
**SUCROSE-CIE**	84.1 ± 25.2	90.1 ± 26.7	66.0 ± 24.5

The data are expressed as mean ± SEM. *n* = 7–8/group.

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
