# Peer review of "Ethanol Reinforcement Elicits Novel Response Inhibition Behavior in a Rat Model of Ethanol Dependence"

_brainsci, 2018, doi:10.3390/brainsci8070119_

Round 1

Reviewer 1 Report

The manuscript from Somkuwar et al reports the consequences of maintaining rats on a schedule of intermittent ethanol exposure by inhalation on performance in two operant tasks. Performance in one task, the DRL paradigm, may under some circumstances reflect impulsive responding, but is open to other interpretations. The title is rather too strong for the methods and data presented here.

There are a number of weaknesses in methodology and interpretation that severely restrict my enthusiasm for the paper. In particular, the use of quite different measures to study performance for ethanol vs. sucrose reinforcers makes any comparison of the two meaningless. Secondly, the analysis and interpretation of the DRL data are difficult to follow.

Given the weaknesses outlined, interpretation of the data is in any case difficult. However, the authors make further quite claims, that are not only speculative, but quite unjustified.

Below are some detailed comments:

Lines 90-91. The authors introduce the idea that performance under DRL may provide a novel measure of increased motivation. However, it is well known that Increased motivation may lead to impaired  response inhibition, and thus may affect DRL performance.

Line 110. What was the operant response requirement? For ethanol DRL performance it is responding on a retractable lever (is the retractable important?). From line 156-160 it seems that the sucrose pellet operant was imply entry into the food chamber. I would question whether this latter behavior Is truly operant, but the use of different procedures in any case makes any comparison inappropriate.

Line 217 et seq. I am not familiar with the approach to analysis of DRL performance. Not all the variables appear to have been defined. E.g. d, N, c, e.

However, I have considerable difficulty in following the authors’ approach. Burst responses are always problematic in DRL analysis, and it is difficult to arrive at a non-arbitrary means to separate them from interval performance of interest.  Here I am not clear how the authors define “timed” and “non-timed” IRTs. Inspection of Fig 3 suggests that a possible approach would be simply to define IRT values less than 5 sec as burst responses, and from Fig 3c I deduce that this is what the authors have done, though this does not appear to be stated anywhere. While this approach would have the danger of excluding a few responses from the analysis of “timed” IRTs, these would have little influence on the overall estimate of performance. If this is what the authors have done, they should simply say so. I think this is the value the authors give as the value q. As far as I can follow the authors, timed IRTs are then simply the total number of responses minus the burst responses. The so-called response threshold, theta, is the mean timed IRT, while it is assumed that a mean IRT less than the required 15sec is a measure of reduced (Impaired?) accuracy. However, it needs to be considered that IRT values greater than 15 are also measures of inaccuracy. Now things get really confusing: Reduced capacity for response inhibition is defined as the fraction of all IRTs represented by timed IRTs. Thus, capacity for response inhibition is assessed by dividing the sum of both IRTs that are premature, and those that are late, by a value that includes burst responding, a quite different measure. Would it not be simpler to analyze the pattern of responding after the burst responding is omitted, simply as a shift in the peak (and other parameters) of the normal distribution? It is the shift in that peak rather than in numbers of burst responses that I would want to use as a measure of degree of premature responding and by interpretation, impulsivity akin the premature responding in the 5-choice task. Fig 1b seems to suggest such a shift in the peak, with the ethanol-CIE group perhaps showing a slight shift to longer intervals. You might look at Stephens and Voet. Behav Pharmacol. 1994 5:4-14 for a way of dealing with such data.

Line 265: “No comparisons were conducted between different types of reinforcers.”  As already mentioned, such a comparison would be impossible as different operant requirements were used for sucrose pellets (simple nose-poking in the food compartment; is this truly operant in the sense of an arbitrary motor response, in contrast to simple approach to the food chamber?) and for ethanol (lever-pressing at a location different from reinforcer delivery). This difference precludes not only a formal statistical comparison, but also informal comparisons between the two behaviors.

Line 323 et seq. It is inappropriate to make comparisons, even informal ones, between the sucrose and ethanol data as the methods used were substantially different.

Line 338-339 Remember the quite different response requirements.

Line 377 et seq. The most impressive data in the manuscript are those from the DRL analysis, with more rapid responding in the CIE group at the beginning of the session. Break points are described and defined in the Methods section, lines 192-200, but are not presented in the Results section. I presume, since plateau numbers of responses were similar, break point may also have been, though inspection of Fig 5 suggests that the CIE animals may have stopped responding at about 100 mins, and while the control group went on for longer, there was a pause for about 20 mins after 20 mins. Using the 15 min definition of breakpoint, was this significant?  Further, the establishment of 15min without responding as the breakpoint seems arbitrary.  Why was this time used?

Line 402. This statement is misleading as quite different methods were used for ethanol and the sucrose reinforcer.

Line 422. “Cognitive load was enhanced”.  What does this mean? Is this simply the increase in the DRL requirement. If so, say so.

Line 440. Again, inappropriate comparison of sucrose and ethanol experiments.

Line 465: Was Breakpoint reported?

471 et seq.  There is no measure of compulsive behavior in these studies, and it is therefore inappropriate to suggest that the ETHANOL-CIE rats exhibited compulsive ethanol-seeking behavior. The two pieces of evidence cited (greater ethanol intake under DRL and steeper rise in PR performance, might be evidence of increased motivation for ethanol, but certainly not for compulsive behavior.

Author Response

We thank the reviewers and the editors for their valuable inputs. We have provided a point-by-point response to the concerns and questions raised by the reviewers below. We have addressed the concerns raised in the revised manuscript; changes are highlighted in yellow. We believe that the manuscript is greatly improved as a consequence of their suggestions.

Reviewer 1

1. The manuscript from Somkuwar et al reports the consequences of maintaining rats on a schedule of intermittent ethanol exposure by inhalation on performance in two operant tasks. Performance in one task, the DRL paradigm, may under some circumstances reflect impulsive responding, but is open to other interpretations. The title is rather too strong for the methods and data presented here.

Response: We have changed the title of this manuscript in response to this critique.

2. There are a number of weaknesses in methodology and interpretation that severely restrict my enthusiasm for the paper. In particular, the use of quite different measures to study performance for ethanol vs. sucrose reinforcers makes any comparison of the two meaningless. Secondly, the analysis and interpretation of the DRL data are difficult to follow. Given the weaknesses outlined, interpretation of the data is in any case difficult. However, the authors make further quite claims, that are not only speculative, but quite unjustified.

Response: Based on this critique, all comparisons between sucrose and ethanol reinforcers have been removed from the revised manuscript. Furthermore, the figures have been rearranged to have separate figures for data from sucrose-reinforced rats and those from ethanol-reinforced rats. The data analysis section has been extensively revised to clarify both the analyses conducted. The discussion has also been restructured to both separate sucrose from ethanol and clarify the interpretation of different types of analyses.

Below are some detailed comments:

3. Lines 90-91. The authors introduce the idea that performance under DRL may provide a novel measure of increased motivation. However, it is well known that Increased motivation may lead to impaired response inhibition, and thus may affect DRL performance.

Response: We thank the reviewer for this comment. The reviewer is correct that “Increased motivation may lead to impaired response inhibition, and thus may affect DRL performance” and this information has been included in the revised Discussion and Abstract. The novelty of the current results is that increased motivation was associated with improved (not impaired) response inhibition.

4. Line 110. What was the operant response requirement? For ethanol DRL performance it is responding on a retractable lever (is the retractable important?). From line 156-160 it seems that the sucrose pellet operant was imply entry into the food chamber. I would question whether this latter behavior Is truly operant, but the use of different procedures in any case makes any comparison inappropriate.

Response: For the statements in “Materials and Apparatus”, the response requirement was not presented since this section is included for the sole purpose of describing in details the conditioning modules used for the experiments. The details for the response requirements are provided in section “2.5 Differential Reinforcement of Low-Rate of Responding (DRL)”.

The retractable lever is the kind of lever used in these chambers, and the levers were extended at the start of each session and stayed extended till the end of the session. This information has been included in the revised methods section “2.5 Differential Reinforcement of Low-Rate of Responding (DRL)”.

Responding for sucrose pellets in our experiments would be considered to be “operant” because responding was strengthened by reinforcement. Under no conditions reported herein, were sucrose pellets available without beam-break produced by the nose-poke. We do acknowledge the criticism that the type of response required to elicit reinforcement was different between the Ethanol and Sucrose reinforced DRL (in the revised Data Analysis section). This issue has been resolved as described in response to Question 2 from Reviewer 1.

5. Line 217 et seq. I am not familiar with the approach to analysis of DRL performance. Not all the variables appear to have been defined. E.g. d, N, c, e.

However, I have considerable difficulty in following the authors’ approach. Burst responses are always problematic in DRL analysis, and it is difficult to arrive at a non-arbitrary means to separate them from interval performance of interest.  Here I am not clear how the authors define “timed” and “non-timed” IRTs. Inspection of Fig 3 suggests that a possible approach would be simply to define IRT values less than 5 sec as burst responses, and from Fig 3c I deduce that this is what the authors have done, though this does not appear to be stated anywhere. While this approach would have the danger of excluding a few responses from the analysis of “timed” IRTs, these would have little influence on the overall estimate of performance. If this is what the authors have done, they should simply say so. I think this is the value the authors give as the value q. As far as I can follow the authors, timed IRTs are then simply the total number of responses minus the burst responses. The so-called response threshold, theta, is the mean timed IRT, while it is assumed that a mean IRT less than the required 15sec is a measure of reduced (Impaired?) accuracy. However, it needs to be considered that IRT values greater than 15 are also measures of inaccuracy. Now things get really confusing: Reduced capacity for response inhibition is defined as the fraction of all IRTs represented by timed IRTs. Thus, capacity for response inhibition is assessed by dividing the sum of both IRTs that are premature, and those that are late, by a value that includes burst responding, a quite different measure. Would it not be simpler to analyze the pattern of responding after the burst responding is omitted, simply as a shift in the peak (and other parameters) of the normal distribution? It is the shift in that peak rather than in numbers of burst responses that I would want to use as a measure of degree of premature responding and by interpretation, impulsivity akin the premature responding in the 5-choice task. Fig 1b seems to suggest such a shift in the peak, with the ethanol-CIE group perhaps showing a slight shift to longer intervals. You might look at Stephens and Voet. Behav Pharmacol. 1994 5:4-14 for a way of dealing with such data.

Response: We have acknowledged in our revised Methods section, that there are several different methods for analyzing IRT data and provided more details regarding the specific parameters in mathematical modeling method that we utilized. Furthermore, data analysis using the arbitrarily implemented cut-off of 5 sec for burst responding has been completely removed from the manuscript to disambiguate the IRT data analysis.

6. A. Line 265: “No comparisons were conducted between different types of reinforcers.”  As already mentioned, such a comparison would be impossible as different operant requirements were used for sucrose pellets (simple nose-poking in the food compartment; is this truly operant in the sense of an arbitrary motor response, in contrast to simple approach to the food chamber?) and for ethanol (lever-pressing at a location different from reinforcer delivery). This difference precludes not only a formal statistical comparison, but also informal comparisons between the two behaviors.

B.   Line 323 et seq. It is inappropriate to make comparisons, even informal ones, between the sucrose and ethanol data as the methods used were substantially different.

C.  Line 338-339 Remember the quite different response requirements.

Response: The response for this issue has been described under Question 2 from Reviewer 1.

7. Line 377 et seq. The most impressive data in the manuscript are those from the DRL analysis, with more rapid responding in the CIE group at the beginning of the session. Break points are described and defined in the Methods section, lines 192-200, but are not presented in the Results section. I presume, since plateau numbers of responses were similar, break point may also have been, though inspection of Fig 5 suggests that the CIE animals may have stopped responding at about 100 mins, and while the control group went on for longer, there was a pause for about 20 mins after 20 mins. Using the 15 min definition of breakpoint, was this significant?  Further, the establishment of 15min without responding as the breakpoint seems arbitrary.  Why was this time used?

Response: Table 5 (now Table 4) in the submitted version as well as the revised manuscript reports the breakpoint data. The Reviewer is correct about the absence of differences between groups in breakpoint.

The 15 min without responding protocol used here has been published by others evaluating drugs of abuse (reviewed (Richardson & Roberts, 1996)). More citations have been included in the revised Methods section to show the specific studies that led to the selection of the method used.

The cumulative analysis for the control rats did not show a “pause for about 20 mins after 20 mins” although it might appear that way on a superficial glance of the graph. Any pause greater than 15 min would have resulted in the programmed consequence of ending the PR session, and no further responses would have been obtained. Closer inspection of the section discussed by the Reviewer shows that perhaps there was a close-to-15 min pause, but the control rats resumed responding following that pause. There is no apparent explanation for this pause. This information has been included in the revised discussion.

8. Line 402. This statement is misleading as quite different methods were used for ethanol and the sucrose reinforcer.

Response: This statement in the revised Discussion section is immediately followed by the acknowledgement of the quite different methods used for ethanol and sucrose reinforce.

9. Line 422. “Cognitive load was enhanced”.  What does this mean? Is this simply the increase in the DRL requirement. If so, say so.

Response: This methodological manipulation has been clarified in the revised Discussion section

10.         Line 440. Again, inappropriate comparison of sucrose and ethanol experiments.

Response: The response for this issue has been described under Question 2 from Reviewer 1.

11.         Line 465: Was Breakpoint reported?

Response: Table 5 now Table 4) in the submitted version as well as the revised manuscript reports the breakpoint data

12.         471 et seq.  There is no measure of compulsive behavior in these studies, and it is therefore inappropriate to suggest that the ETHANOL-CIE rats exhibited compulsive ethanol-seeking behavior. The two pieces of evidence cited (greater ethanol intake under DRL and steeper rise in PR performance, might be evidence of increased motivation for ethanol, but certainly not for compulsive behavior.

Response: The phrase “compulsive ethanol-seeking behavior” has been removed from the revised manuscript.

Reviewer 2 Report

A manuscript from Somkuwar assesses the effects of 6 weeks of chronic intermittent ethanol (CIE) exposure via vapor inhalation on the DRL performance of rats.  They find that CIE affects DRL when the reinforcer is supersaccharin sweetened ethanol, but not when it is sucrose pellets, although the effect is comparatively modest and late-emerging.  Contrary to their hypothesis, ethanol vapor exposure decreases, rather than increases observed impulsivity in terms of efficiency and temporal accuracy of responding, but only when sweetened ethanol is a reinforcer. This is a reasonably interesting finding, as increases in impulsivity are thought to underlie excessive drinking, which can occur following alcohol vapor exposure.  On the whole, the paper is appropriately analyzed and presented, and the detailed analysis of behavior during DRL is a real strength.  There are, however, some issues that should be addressed in future revisions, especially regarding the interpretation of these findings.

1.    As a discussion point, authors should consider the possibility that alcohol vapor exposure alters sensitivity to alcohol, and that this change in sensitivity in turn affects DRL responding.  This would explain why alcohol-, but not sucrose-reinforced responding is affected by vapor exposure.  During the DRL (but not progressive ratio) sessions, rats are consuming what may be a pharmacologically relevant amount of alcohol, although unfortunately blood alcohol levels weren’t taken.  Intakes were 1 to (in the case of CIE rats) 1.5 g/kg in a 30-min session.  These fairly substantial doses can either increase or decrease operant response rates, which can in turn affect DRL performance; furthermore, rats can acquire and lose tolerance to these effects (Bird and Holloway, 1989). The authors need to point out that when alcohol is used as a reinforcer, DRL responding is not a “pure” measure of impulsivity, but rather some combination of acute alcohol effects on response rate along with its effects on timing and impulse control.  On the other hand, the sucrose responding, which was unaffected by CIE exposure, is not subject to the confound of intoxication during the session, and readers should understand this.

2.    Although the authors acknowledge that their observed effects are opposite their hypothesis, they don’t really actually consider what this might mean.  They instead argue that during ethanol reinforcement the rats are tapping into some kind of compulsivity circuit, a discussion that seems entirely beside the point (see, for example, point 4).  What if it instead means that impulsivity, as measured by DRL, in fact doesn’t underlie escalating drinking following CIE?  What, then, would be the take-home message?  Or what if decreases in impulsivity caused by CIE actually underlie increases in drinking?  In other words, the authors should discuss the data they have in the context of their hypothesis as written, rather than changing the hypothesis after the fact to fit a puzzling result.

3.    Systematic presentation of effect sizes would aid understanding of these changes in behavior. Changes in DRL measures are quite small; even the authors indicate that they are “small but statistically significant.” But there is no real discussion about whether they are behaviorally meaningful.  While this may be in the eye of the beholder, presentation of effect sizes is both good science and a place to start the discussion of whether these findings are meaningful.  Furthermore, there is always the concern that these findings are significant but not replicable, especially given their modesty.  Would the authors have expected DRL differences to emerge so long after CIE exposure, but not be apparent during the CIE period?  If so, why?

4.    Some discussion section points should be stricken.  One, on lines 443-445, was particularly problematic, stating “assuming control rats were responding at an optimal rate…”  To say that rats that were more impulsive (air control) were somehow optimal simply because they had no history of alcohol exposure is to set up an irrefutable hypothesis that alcohol exposure is ALWAYS bad.  To go on and talk about “compulsive alcohol intake” in these animals when they were in fact more able to INHIBIT their alcohol seeking behavior shows sloppy thinking.  The more likely, and certainly more simple explanation is that alcohol exposed rats were better able to deal with alcohol intoxication at test, and were, in fact, more optimal in their behavior!  These discussion ideas need to be reexamined, and the section as a whole is somewhat disorganized and could use rewriting and streamlining.  

5.    Authors should tone down their conclusions that CIE rats showed greater alcohol avidity during progressive ratio.  The fact remains that for the main dependent variable, break point, there was no difference; the pattern of responding during the session is a much less salient, secondary measure.  Does the lack of differences in breakpoint somewhat refute the predicted result for protracted abstinence and ethanol reinforcing efficacy?

6.    Authors should acknowledge in the text of the paper that they greatly missed their target BECs during week 1, and discuss the implications, if any, of this fact. Furthermore, they should assess whether sucrose and ethanol responding rats differed in BECs at any time during CIE using a Group X Week ANOVA on BECs.

7.    A closer inspection for spelling errors would be helpful (e.g., ethnaol in Fig 1, saccharine in Discussion).

Author Response

Reviewer 2

A manuscript from Somkuwar assesses the effects of 6 weeks of chronic intermittent ethanol (CIE) exposure via vapor inhalation on the DRL performance of rats.  They find that CIE affects DRL when the reinforcer is supersaccharin sweetened ethanol, but not when it is sucrose pellets, although the effect is comparatively modest and late-emerging.  Contrary to their hypothesis, ethanol vapor exposure decreases, rather than increases observed impulsivity in terms of efficiency and temporal accuracy of responding, but only when sweetened ethanol is a reinforcer. This is a reasonably interesting finding, as increases in impulsivity are thought to underlie excessive drinking, which can occur following alcohol vapor exposure.  On the whole, the paper is appropriately analyzed and presented, and the detailed analysis of behavior during DRL is a real strength.  There are, however, some issues that should be addressed in future revisions, especially regarding the interpretation of these findings.

1.    As a discussion point, authors should consider the possibility that alcohol vapor exposure alters sensitivity to alcohol, and that this change in sensitivity in turn affects DRL responding.  This would explain why alcohol-, but not sucrose-reinforced responding is affected by vapor exposure.  During the DRL (but not progressive ratio) sessions, rats are consuming what may be a pharmacologically relevant amount of alcohol, although unfortunately blood alcohol levels weren’t taken.  Intakes were 1 to (in the case of CIE rats) 1.5 g/kg in a 30-min session.  These fairly substantial doses can either increase or decrease operant response rates, which can in turn affect DRL performance; furthermore, rats can acquire and lose tolerance to these effects (Bird and Holloway, 1989). The authors need to point out that when alcohol is used as a reinforcer, DRL responding is not a “pure” measure of impulsivity, but rather some combination of acute alcohol effects on response rate along with its effects on timing and impulse control.  On the other hand, the sucrose responding, which was unaffected by CIE exposure, is not subject to the confound of intoxication during the session, and readers should understand this.

Response: We thank the reviewer for this comment and have included in the revised Discussion that tolerance to alcohols effects in the CIE rats may contribute to the differences in responding under the DRL schedule. However, the rats are consuming 1-1.5 mg/kg during the 30-min session and not 1-1.5 g/kg, which is about three orders of magnitude lower than that observed under fixed ratio schedule.

2.    Although the authors acknowledge that their observed effects are opposite their hypothesis, they don’t really actually consider what this might mean.  They instead argue that during ethanol reinforcement the rats are tapping into some kind of compulsivity circuit, a discussion that seems entirely beside the point (see, for example, point 4).  What if it instead means that impulsivity, as measured by DRL, in fact doesn’t underlie escalating drinking following CIE?  What, then, would be the take-home message?  Or what if decreases in impulsivity caused by CIE actually underlie increases in drinking?  In other words, the authors should discuss the data they have in the context of their hypothesis as written, rather than changing the hypothesis after the fact to fit a puzzling result.

Response: We have modified the Discussion section to discuss the potential implications of decreased impulsivity in contributing to addiction-like phenotype in CIE rats.

3.    Systematic presentation of effect sizes would aid understanding of these changes in behavior. Changes in DRL measures are quite small; even the authors indicate that they are “small but statistically significant.” But there is no real discussion about whether they are behaviorally meaningful.  While this may be in the eye of the beholder, presentation of effect sizes is both good science and a place to start the discussion of whether these findings are meaningful.  Furthermore, there is always the concern that these findings are significant but not replicable, especially given their modesty.  Would the authors have expected DRL differences to emerge so long after CIE exposure, but not be apparent during the CIE period?  If so, why?

Response: The effect size for the increase impulsivity is presented in the revised Results section

4.    Some discussion section points should be stricken.  One, on lines 443-445, was particularly problematic, stating “assuming control rats were responding at an optimal rate…”  To say that rats that were more impulsive (air control) were somehow optimal simply because they had no history of alcohol exposure is to set up an irrefutable hypothesis that alcohol exposure is ALWAYS bad.  To go on and talk about “compulsive alcohol intake” in these animals when they were in fact more able to INHIBIT their alcohol seeking behavior shows sloppy thinking.  The more likely, and certainly more simple explanation is that alcohol exposed rats were better able to deal with alcohol intoxication at test, and were, in fact, more optimal in their behavior!  These discussion ideas need to be reexamined, and the section as a whole is somewhat disorganized and could use rewriting and streamlining. 

Response: This phrase has been removed from the revised Discussion

5.    Authors should tone down their conclusions that CIE rats showed greater alcohol avidity during progressive ratio.  The fact remains that for the main dependent variable, break point, there was no difference; the pattern of responding during the session is a much less salient, secondary measure.  Does the lack of differences in breakpoint somewhat refute the predicted result for protracted abstinence and ethanol reinforcing efficacy?

Response: This portion of the discussion has been modified to accommodate the Reviewer’s concern.

6.    Authors should acknowledge in the text of the paper that they greatly missed their target BECs during week 1, and discuss the implications, if any, of this fact. Furthermore, they should assess whether sucrose and ethanol responding rats differed in BECs at any time during CIE using a Group X Week ANOVA on BECs.

Response: The low BEC for week 1 has been included in the revised Methods and Results section since this was an expected finding. The objective of BECs is to determine how to titrate the flow properties of air and ethanol vapors. Evaluating the effects of BEC on impulse control behavior is a very interesting and important question that should be evaluated in future studies. It is unlikely that week-to-week variabilities in  BECs affected behavior reported herein since both efficiency and Intake of SUCROSE as well as ETHANOL show fairly steady values across the weeks of CIE (Fig 1 and 3).

7.    A closer inspection for spelling errors would be helpful (e.g., ethnaol in Fig 1, saccharine in Discussion).

Response: We thank the reviewer for catching these typographical errors. We have carefully inspected the revised manuscript to correct such errors prior to resubmission.

Round 2

Reviewer 2 Report

The authors have revised the paper, but questions still remain, which I enumerate below.

1) Earlier I had asked the authors to consider and discuss the fact that differential sensitivity to alcohol could potentially explain differences in DRL behavior between CIE and control groups.  I pointed out that the amount of alcohol consumed in DRL was substantial, about 1-1.5 g/kg ethanol, and that differences in sensitivity to the depressant effects of ethanol could explain group differences in DRL behavior when ethanol is the reinforcer.  In response, the authors pointed out that the rats consumed not 1-1.5 g/kg ethanol, but merely 1.5 MG/KG ethanol, three orders of magnitude lower than during FR responding.  This is a tiny amount, and would mean that the rats could consume no more than about 0.01 ml 10% ethanol if my calculations are correct.  (1.5 mg/kg ethanol in a half-kg rat is 0.75 mg ethanol, which at 10% (v/v) is 9.5 mg of 10% ethanol, or 0.01 ml).  This seems impossible, but in rereading the manuscript, I could find nowhere that the authors state the volume of ethanol delivered per reinforcer during DRL, nor the number of reinforcers consumed.  These are critical pieces of information that need to be provided in a follow-up.  However, the response raises more important questions about the rats' behavior during DRL that need to be answered.  I continue hoping it is an error that rats only consumed 1.5 mg/kg ethanol during DRL, but if not, some serious reconsideration of experimental procedures is in order.  Alternatively, if the rats were consuming a pharmacologically relevant amount of ethanol during DRL, the discussion needs to consider differential sensitivity to ethanol as a potential confound/explanatory factor that might be more plausible than stating (as the current Discussion does) that increased motivation leads to lower impulsivity, and that lower impulsivity is a consequence of ethanol withdrawal.

2) In the earlier version, I asked the authors to explicitly analyze whether BECs during CIE were different between sucrose and ethanol groups.  For some reason, they decided not to provide this analysis, which seems odd.  If they don't want to add it to the paper for some reason, perhaps they could provide the analysis in the rebuttal letter only?

3) Saccharine is still misspelled in the text, as is Ethnaol in Figure 3, notwithstanding my earlier comment.

Author Response

We thank Reviewer 2 for the additional comments and input. We provide detailed responses below. We addressed the concerns in the revised manuscript. Changes are highlighted in yellow. We believe that the comments greatly improved the manuscript.

Reviewer 2

1) Earlier I had asked the authors to consider and discuss the fact that differential sensitivity to alcohol could potentially explain differences in DRL behavior between CIE and control groups.  I pointed out that the amount of alcohol consumed in DRL was substantial, about 1-1.5 g/kg ethanol, and that differences in sensitivity to the depressant effects of ethanol could explain group differences in DRL behavior when ethanol is the reinforcer.  In response, the authors pointed out that the rats consumed not 1-1.5 g/kg ethanol, but merely 1.5 MG/KG ethanol, three orders of magnitude lower than during FR responding.  This is a tiny amount, and would mean that the rats could consume no more than about 0.01 ml 10% ethanol if my calculations are correct.  (1.5 mg/kg ethanol in a half-kg rat is 0.75 mg ethanol, which at 10% (v/v) is 9.5 mg of 10% ethanol, or 0.01 ml).  This seems impossible, but in rereading the manuscript, I could find nowhere that the authors state the volume of ethanol delivered per reinforcer during DRL, nor the number of reinforcers consumed.  These are critical pieces of information that need to be provided in a follow-up.  However, the response raises more important questions about the rats' behavior during DRL that need to be answered.  I continue hoping it is an error that rats only consumed 1.5 mg/kg ethanol during DRL, but if not, some serious reconsideration of experimental procedures is in order. 

Response: We apologize for providing an incorrect response in our last response to the reviewer’s comments with regard to the amount of ethanol that the rats consumed. We also corrected the reported units. We also included the number of rewards during the session in addition to the g/kg consumed in the revised Fig. 3. We provided the volume of ethanol delivered for each reward in the Fig. 3 legend.

2) Alternatively, if the rats were consuming a pharmacologically relevant amount of ethanol during DRL, the discussion needs to consider differential sensitivity to ethanol as a potential confound/explanatory factor that might be more plausible than stating (as the current Discussion does) that increased motivation leads to lower impulsivity, and that lower impulsivity is a consequence of ethanol withdrawal.

Response: We revised the Discussion to address the reviewer’s comments (see Page 12-13).

3) In the earlier version, I asked the authors to explicitly analyze whether BECs during CIE were different between sucrose and ethanol groups.  For some reason, they decided not to provide this analysis, which seems odd.  If they don't want to add it to the paper for some reason, perhaps they could provide the analysis in the rebuttal letter only?

Response: We included the BECs for the sucrose and ethanol groups in the revised Fig. 1 and 3. We did not compare BECs between these two groups in the revised manuscript based on Reviewer 1’s comments, which indicated that the two groups needed to be analyzed separately. Below is the analysis, and we defer to the editor whether to include this information in the manuscript itself.

Analysis: BECs for week 5 in the Ethanol-CIE group were not available because of technical challenges that are reported in the manuscript. Thus, BECs for Week 5 in both groups had to be excluded to conduct the two-way ANOVA. Significant main effects of time and treatment on BECs during CIE were observed (Ftime[4, 67] = 63.8, p < 0.0001; FReinforcement[1, 67] = 22.5, p < 0.0001; Finteraction[4, 67] = 19.6, p < 0.0001). Sidak’s post hoc test revealed that BECs were different between groups during Weeks 1, 3, and 4 and not different between groups during Weeks 2 and 6.

3) Saccharine is still misspelled in the text, as is Ethnaol in Figure 3, notwithstanding my earlier comment.

Response: We thank the reviewer for bringing this to our attention. The revised manuscript was thoroughly proofread.
